# Pristine and Poly(Dimethylsiloxane) Modified Multi-Walled Carbon Nanotubes as Supports for Lipase Immobilization

**DOI:** 10.3390/ma14112874

**Published:** 2021-05-27

**Authors:** Iryna Sulym, Jakub Zdarta, Filip Ciesielczyk, Dariusz Sternik, Anna Derylo-Marczewska, Teofil Jesionowski

**Affiliations:** 1Chuiko Institute of Surface Chemistry of NASU, 17 General Naumov Str., 03164 Kyiv, Ukraine; 2Institute of Chemical Technology and Engineering, Faculty of Chemical Technology, Poznan University of Technology, Berdychowo 4, 60965 Poznan, Poland; Filip.Ciesielczyk@put.poznan.pl (F.C.); teofil.jesionowski@put.poznan.pl (T.J.); 3Institute of Chemical Sciences, Maria Curie-Sklodowska University in Lublin, M.C. Sklodowska Sq.3, 20031 Lublin, Poland; dsternik@poczta.umcs.lublin.pl (D.S.); annad@hektor.umcs.lublin.pl (A.D.-M.)

**Keywords:** multi-walled carbon nanotubes, poly(dimethylsiloxane), polymer nanocomposites, *Candida antarctica* lipase B, lipase immobilization, enzymes stability and reusability

## Abstract

The presented study deals with the fabrication of highly stable and active nanobiocatalysts based on *Candida antarctica* lipase B (CALB) immobilization onto pristine and poly(dimethylsiloxane) modified MWCNTs. The MWCNTs/PDMS nanocomposites, containing 40 wt.% of the polymer with two molecular weights, were successfully synthesized via adsorption modification. The effect of PDMS chains length on the textural/structural properties of produced materials was studied by means of the nitrogen adsorption–desorption technique, Raman spectroscopy, and attenuated total reflectance Fourier transform infrared spectroscopy. P-MWCNTs and MWCNTs/PDMS nanocomposites were tested as supports for lipase immobilization. Successful deposition of the enzyme onto the surface of P-MWCNTs and MWCNTs/PDMS nanocomposite materials was confirmed mainly using ATR-FTIR spectroscopy. The immobilization efficiency, stability, and catalytic activity of the immobilized enzyme were studied, and the reusability of the produced biocatalytic systems was examined. The presented results demonstrate that the produced novel biocatalysts might be considered as promising materials for biocatalytic applications.

## 1. Introduction

Carbon occurs in many forms, and the properties of each form with respect to their special structure make carbon a truly unique building block for nanomaterials [1]. Owing to their interesting electrical, magnetic, mechanical, and thermal properties, carbon nanotubes (CNTs) have become the most promising materials in many scientific and technological fields [2,3,4,5,6,7]. CNTs are made of one or more graphene sheets rolled-up to form tubes. Single-walled CNTs (SWCNTs) comprise a single graphene layer seamlessly wrapped into a cylindrical tube. Multi-walled carbon nanotubes (MWCNTs) comprise an array of concentric cylinders coaxially arranged around a central hollow core with van der Waals forces between adjacent layer [8,9]. CNTs exhibit extremely high surface area, large aspect ratio, low density, remarkably high mechanical strength, and electrical as well thermal conductivities [8,10]. Functionalization of carbon nanotube surfaces can be proceeded viacovalent or non-covalent modification. Non-covalent surface modificationof CNTs, which includes adsorption of surfactants, polymers, or biological macromolecules, is a method that does not affect their intrinsic structure [11]. Among many methods, non-covalent attachment (polymer wrapping and absorption) is of key importance when CNTs functionalize with polymer molecules [11,12,13]. Polymeric materials includingsilicon rubbers, in particular poly(dimethylsiloxane) (PDMS), are used in everyday materials. It is related to their excellent durability and mechanical properties that results from their high cross-linking density and degree of polymerization. PDMS is hydrophobic, chemically inert, and electrically resistant, and exhibits dielectric strength, high elasticity, and easy processing and isenvironmentally-friendly [14,15,16,17]. The addition of PDMS to MWCNTs improves the mechanical, electrical, and thermal properties of the resulting nanocomposites [18,19,20]. Based on our previous studies [21,22,23,24,25] concerning materials prepared in a similar way but containing chemically different highly dispersed nano-oxides as well as PDMS modified MWCNTs, it was proved thatpolymer–filler interactions depend on nanocomposite compositions and inorganic particle types (oxides). On the other hand, those interactions are responsible for surface properties of the resulting nanocomposites such as roughness, textural porosity, or hydrophobicity, which furthermore affect their potential applications as catalysts supports, adsorbents, fillers, etc.

This current research may be a part of work on preparation of active biocatalyst composed with the functional support and biomolecules—e.g., enzymes.

Recently, many studies have been devoted to the immobilization of enzymatically active substances on inorganic, e.g., silica-based, supports towards heterogeneous biocatalyst preparation. This is especially related to the nature of such systems—their availability and relatively low cost as well as high mechanical strength and durability of biocatalyst granules in the reaction media [26,27]. In particular, lipases are the most widely used biomolecules in enzyme technology because of their widespecificity for some substrates, low production cost, wide pH activity profiles, as well as ability to catalyze various different reactions, such as hydrolysis, esterification, amination, transesterification, alcoholysis, etc. [28,29,30]. Graphene oxide (GO), having a large surface area (2630 m^2^/g) and abundant functional groups (such as epoxide, hydroxyl, and carboxylic groups), provides a great substrate for enzyme immobilization without any surface modification or any coupling agents [31]. Nevertheless, among the various nanostructured materials that might be used as novel supports for enzyme immobilization and stabilization, CNTs are of great interest to many research centers worldwide due to their stability, high adsorption capacity, improvedretention of catalytic activity, and biocompatibility [32,33,34,35]. Both SWCNTs and MWCNTs have been used to immobilize enzymes [36]. MWCNTs are structurally similar to SWCNTs, but their diameters can range from a few nanometers to dozens of nanometers [37]. SWCNTs are attractive due to their larger surface area for protein interaction, excellent biocompatibility, antifouling properties, and high conductivity, but MWNTs are desirable because they are easier to prepare, exhibit better dispersibility, and are commercially available at a relative lower price, which makes them more feasible for industrial applications. Therefore, MWNTs are suitable materials as enzyme supports [38]. Enzymes can be immobilized on the surface of MWCNTs by adsorption or covalent binding, whichresultsin enhanced catalytic performance and stability. Moreover, lipases are well-known interfacially active catalysts and exhibit their catalytic abilities at the interface between the organic phase containing hydrophobic substrates and aqueous phase, so they can be activated at the hydrophobic–hydrophilic interface [34,39].

Herein, in the present work, for the first time, the methodology to design and characterize an alternative, highly stable, and active nanobiocatalyst based on *Candida antarctica* lipase B (CALB) immobilized onto pristine and poly(dimethylsiloxane)modified MWCNTsis presented and discussed. The idea was to combine the textural properties as well as functionality of PDMS modified with lipase activity, and to obtain a novel type of biocatalyst dedicated to biotechnological applications. As a result, high enzyme loading, its improved stability and reusability, as well as activity of the biocatalyst produced, were expected. The innovative nature of the presented study is based not only on application of a novel, previously undescribed support material for lipase immobilization, but also on the possibility to use MWCNT surface modifying agents at various molecular weights to examine their effects on enzyme loading and catalytic activity.

## 2. Materials and Methods

### 2.1. Chemicals and Materials

Commercial poly(dimethylsiloxane) fluids of two molecular weights (Wacker Chemie AG, linear, –CH_3_ terminated, code names: PDMS-100 and PDMS-12500 with MW ≈ 3410 and 39,500 g/mol, respectively) and multi-walled carbon nanotubes (MWCNTs, obtained by catalytic chemical vapor deposition (CCVD) [40] using pyrolysis of propylene on complex metal oxide catalysts) [41] were used as initialmaterials forpolymer nanocompositepreparation.

*Candida antarctica* lipase B (CALB) (EC 3.1.1.3, ~200 U/g), para-nitrophenyl palmitate (p-NPP), para-nitrophenol (p-NP), gum arabic and Triton X-100 (laboratory grade), sodium hydroxide, sodium chloride, 50 mM acetate buffer (pH 3–5), 50 mM phosphate buffer (pH 6–8), and 50 mM Tris-HCl (pH 8 and 9) were supplied from Sigma-Aldrich (St. Louis, MO, USA). All of the reagents were of analytical grade and were used directly without any further purification.

### 2.2. Preparation of MWCNTs/PDMS Nanocomposites

PDMS-100 and PDMS-12500 fluids were physically adsorbed onto pristine multi-walled carbon nanotubes (P-MWCNTs) in the amount of 40 wt.%. Before adsorption, the samples were dried at 110 °C for 2 h. A hexane solution of PDMS (1 wt.% PDMS) was prepared, and its estimated amount wasadded to a fixed amount of dry carbon powder material. The suspension was mechanically stirred and finally dried at room temperature for 48 h and then at 80 °C for 3 h. All samples modified with PDMS in the amount of 40 wt.% were in the form of powder similar to P-MWCNTs, while neat PDMS-100 and PDMS-12500 were liquids. The prepared polymer nanocomposites were marked as MWCNTs/PDMS-100(40) and MWCNTs/PDMS-12500(40), respectively.

### 2.3. Lipase Immobilization

The pristine MWCNTs and MWCNTs/PDMS nanocomposites were used as supports for the immobilization of *Candida antarctica* lipase B (CALB). In all experiments, 100 mg of P-MWCNTs or modified MWCNTs (MWCNTs/PDMS-100(40) and MWCNTs/PDMS-12500(40)) were added to 5 mL of lipase solution at concentration of 5 mg/mL in 50 mM phosphate buffer solution at pH 7. The samples were then shaken for 24 h using a KS 4000i Control incubator (IKA Werke GmbH, Staufen im Breisgau, Germany) at ambient temperature. Next, samples were centrifuged (4000 rpm at 4 °C) using an Eppendorf 5810 R centrifuge (Hamburg, Germany) and furthermore washed several times with 50 mM phosphate buffer in order to remove unbounded lipase. The samples were labelled as CALB@P-MWCNTs, CALB@MWCNTs/PDMS-100(40), and CALB@MWCNTs/PDMS-12500(40), respectively.

### 2.4. Analysis of Nanocomposites before Lipase Immobilization

#### 2.4.1. Textural Characterization

To analyze the textural characteristics of P-MWCNTs and MWCNTs/PDMS nanocomposites, low-temperature (77.4 K) nitrogen adsorption–desorption isotherms were recorded using an automatic gas adsorption analyzer ASAP 2420 (Micromeritics Instrument Corp., Norcross, GA, USA). Beforehand, the measurement samples were degassed at 110 °C for 2 h in a vacuum chamber. The values of surface area (*S*_BET_) were calculated according to the standard BET method (using Micromeritics software). The total pore volume, *V*_p_, was evaluated from the nitrogen adsorption at *p*/*p*_0_ = 0.98–0.99 (*p* and *p*_0_ denote the equilibrium and saturation pressure of nitrogen at 77.4 K, respectively). The nitrogen desorption data were used to compute the pore size distributions (PSD, differential *f*_V_(R)~d*V*p/d*R* and *f*_S_(R) ~ d*S*/d*R*)), using a model with slit-shaped and cylindrical pores and voids between spherical nanoparticles (SCV) with a self-consistent regularization (SCR) procedure for MWCNTs/PDMS samples and slit-shaped pores for P-MWCNTs [42,43]. The differential PSD with respect to pore volume *f*_V_ ~ d*V*/d*R*, ∫*f*_V_d*R* ~ *V*_p_ were re-calculated to incremental PSD (IPSD) at ϕ_V_(*R_i_*) = (*f*_V_(*R_i_*_+1_) + *f*_V_(*R_i_*))(*R_i_*_+1_ − *R_i_*)/2 at ∑ϕ_V_(*R_i_*) = *V*_p_). The *f*_V_ and *f*_S_ functions were also used to calculate contributions of micropores (*V*_micro_ and *S*_micro_ at radius *R* ≤ 1 nm), mesopores (*V*_meso_ and *S*_meso_ at 1 nm < *R* < 25 nm), and macropores (*V*_macro_ and *S*_macro_ at 25 nm < *R* < 100 nm) to the total pore volume and surface area.

#### 2.4.2. Spectral Analysis

The Raman spectra were recorded over the 3200–500 cm^−1^ range using the in Via Reflex Microscope DMLM Leica Research Grade, Reflex (Renishaw, Wotton-under-Edge, UK) with Ar^+^ ion laser excitation at λ_0_ = 514.5 nm. For each sample, the spectra were recorded at several points in order to ascertain the homogeneity of the sample, and the average results were plotted. The surface functional groups of the pristine MWCNTs and MWCNTs/PDMS nanocomposites, before and after lipase immobilization, were investigated using Fourier transform infrared spectroscopy (FTIR) in attenuated total reflectance (ATR) mode (Vertex 70 spectrometer, Bruker, Germany). The FTIR spectra were recordedat a wavenumber range of 4000–500 cm^−1^, and at a resolution of 1 cm^−1^.

### 2.5. Characterization of Free and Immobilized Lipase

The activity of free and immobilized lipase was estimated based on the model reaction of p-nitrophenyl palmitate hydrolysis to p-nitrophenol. The spectrophotometric measurements at 410 nm, using a Jasco V-750 UV–Vis spectrophotometer (Jasco, Tokyo, Japan), were used for this purpose. In the reaction, 10 mg of free lipase and a corresponding amount of the biocatalytic system produced (CALB@MWCNTs, CALB@MWCNTs/PDMS-100(40) and CALB@MWCNTs/PDMS-12500(40)), containing 10 mg of the enzyme, were used. Reactions were carried out for 2 min under continuous stirring (800 rpm). After the assumedtime, the reaction was terminated, and absorbance was measured. The activity of the free and immobilized lipase was calculated using a standard calibration curve for p-NP. The amount of biocatalyst that hydrolyzed 1 μmol of p-NPP per minute was defined as the one unit of lipase activity. The highest measured activity was defined as 100% relative activity. The effect of pH on the activity of the free and immobilized lipase was determined based on the above-mentioned reaction at a temperature of 30 °C, in the pH range 3–11, using buffer solution at the desired pH. The effect of temperature on the activity of the free and immobilized lipase was determined according to the above-presented methodology at pH 7 (50 mM phosphate buffer), over a temperature rangefrom 10 to 80 °C. All measurements were made in triplicate; error bars are presented as means ± standard deviation.

### 2.6. Stability and Reusability of Free and Immobilized Lipase

Thermal stability over time was determined during incubation of free and immobilized enzyme for 180 min under optimal process conditions (30 °C and pH 7). The relative activity of free and immobilized lipase was further determined based on the model hydrolysis reaction of p-nitrophenyl palmitateat every specified period of time, applying spectrophotometric measurements at 410 nm. The initial lipase activity was defined as 100% relative activity. The inactivation constant (*k_D_*) and enzyme half-life (*t*_1/2_) were determined based on the linear regression slope.

Storage stability of free lipase and products after immobilization were examined based on the above-mentioned model reaction of p-NPP hydrolysis over 30 days of storage at 4 °C in 50 mM phosphate buffer at pH 7. The relative activity was measured every 2 days.

The reusability of the produced biocatalytic systems was also determined using the same model hydrolysis reaction carried out under optimal process conditions, over 10 repeated catalytic cycles. Immobilized lipase was separated from the reaction mixture by centrifugation, washed several timeswith 50 mM phosphate buffer at pH 7, and placed in the fresh substrate solution.

The effect of 5% Triton X-100 and 0.5 M NaCl on relative activity of the immobilized enzyme was examined over 24 h of incubation in a proper solution. After a specified period of time, the relative activity of the immobilized lipase was determined based on the model reaction of p-NPP hydrolysis. All measurements were made in triplicate; error bars are presented as means ± standard deviation.

## 3. Results

### 3.1. Analysis of Nanocomposites before Lipase Immobilization

#### 3.1.1. Parameters of the Porous Structure

The structural characteristic of P-MWCNTs and MWCNTs/PDMS nanocomposites wasstudied using low-temperature nitrogen adsorption–desorption isotherms (Figure 1a and Table 1). All of the materials were characterized withthe nitrogen adsorption isotherms of type II (H3 type of hysteresis loops) according to the IUPAC classification [44,45]. Capillary condensation occurred at pressure *p*/*p*_0_ > 0.85 (due to adsorption in broad mesopores and macropores).

Surface area values (Table 1, *S*_BET_) demonstrated a significant reduction after adsorption of both types of PDMS onto carbon nanotube surfaces. Moreover, the total pore volume (*V*_p_) decreased for the MWCNTs/PDMS-100 and MWCNTs/PDMS-12500 nanocomposites by 68 and 59%, respectively, as compared to the P-MWCNTs. Moreover, it was observed that the pore average radii in MWCNTs/PDMS (−100, −12,500) samples was three times greater than that of P-MWCNTs.

The incremental pore size distribution IPSD functions (Figure 1b) show that the textural characteristics ofMWCNTschanged after the modification with polymer. The textural porosity of the pristine MWCNT resulted from mesopores and secondly due to micropore presence. MWCNTs/PDMS nanocomposites were characterized by bimodal porous structures (Figure 1b). In addition, MWCNTs/PDMS samples were characterized with a significant decrease in mesopore contributions to the total porosity with a simultaneous increase in contributions of macropores as compared to P-MWCNTs.

#### 3.1.2. Raman Spectroscopy

Raman spectroscopy is a very valuable tool for the characterization of carbon-based nanostructures. This technique is used to analyze the presence of amorphous and crystalline phases corresponding to differences in graphitization. The spectra were collected in the most informative range for carbon materials of 3200–500 cm^−1^ (Figure 2). Three major peaks at 1341 cm^−1^ as the D-band (sp^3^ carbons in non-graphitic structures), at 1570 cm^−1^ as the G-band (sp^2^ carbons in graphitic structures), and its second-order harmonic at 2672 cm^−1^ as the G’-band, were noted [46]. The ratio between the integral intensities of the G and D bands (A_G_/A_D_ ratio as a measure of the graphitization degree) is an indicator of the crystallinity degree [47]. The value of A_G_/A_D_ was calculated by deconvolution of the spectra using the Lorentzian function. After adsorption of polymer, relative intensity ratio A_G_/A_D_ tended to decrease from 1.1 for P-MWCNTs to 0.95 for MWCNTs/PDMS nanocomposites, respectively.

#### 3.1.3. ATR-FTIR Spectroscopy

Fourier transform infrared spectroscopy was used to determine the nature of chemical groups present on the surface of analyzed materials as well as to indirectly confirm nanotube modification and enzyme immobilization (Figure 3).

The FTIR spectrum of the P-MWCNTs showeda broad peak with a maximum at 1060 cm^−1^ that corresponded to the stretching vibrations of C–O bonds in carboxylic groups present onto the surface of MWCNTs. Upon modification by PDMS, irrespectively of the molecular mass of the modifying agent used, additional signals could be observed. The small signal at 2950 cm^−1^ was related to the presence of C–H stretching vibrations, peaks at 1250 and 780 cm^−1^ corresponded to the stretching and bending vibrations of Si-CH_3_ groups, whereas peaks at 1010 and 1055 cm^−1^ were characteristic for the stretching vibrations of Si–O–Si bonds [48]. After enzyme immobilization onto both materials, the presence of additional signals, characteristic for the enzyme structure, was clearly seen. Among them, the most important was a peak at 3400 cm^−1^, assigned to the stretching vibrations of –OH groups, and two signals at 1655 and 1545 cm^−1^, ascribed to the stretching vibrations of amide I and amide II bands, respectively. Further, it could be seen that the intensity of signals characteristic for enzyme was higher in the CALB@MWCNTs/PDMS-100(40) spectrum, as compared to the CALB@MWCNTs/PDMS-12500(40) spectrum.

### 3.2. Characterization of Free and Immobilized Lipase

The next stage included tests of obtained materials (MWCNTs modified with 40 wt.% of PDMS-100 and PDMS-12500) as supports for enzyme immobilization. Lipase was selected as a model enzyme, as itexhibitsimproved catalytic activity in a hydrophobic microenvironment. The effect of various process conditions on thestability and activity of the immobilized enzyme was determined, and the reusability of the produced biocatalytic systems was examined.

#### 3.2.1. pH Profiles of Free and Immobilized Lipase

Free lipase and biocatalytic systems produced showed maximum activity at pH 7 (Figure 4). Further, their pH profiles weresimilar. In the tested pH range (beside pH 7), free enzyme exhibited relative activity not higher than 60% and even less than 30% at pH ranges from 3 to 5 and from 9 to 11. By contrast, enzyme immobilized on both pristine and modified MWCNTs showed about 10–30% higher relative activity over whole analyzed pH range. Further, lipase deposited onto MWCNTs/PDMS nanocomposites retained over 80% of its relative activity over a wide pH rangefrom 6 to 9 and more than 30% relative activity at pH 3 to 10. It should also be highlighted that lipase immobilized onto MWCNTs/PDMS-100(40) material exhibited around 5–10% higher activity than enzyme immobilized onto carbon nanotubes functionalized by PDMS withhigher molecular weight.

#### 3.2.2. Temperature Profiles of Free and Immobilized Lipase

Temperature profiles of free and immobilized lipase were determined over a temperature range of 10–80 °C (Figure 5). The optimal temperature for all analyzed samples was found to be 30 °C. Even a slight change in temperature conditions resulted in a sharp decrease of catalytic activity of free enzyme. Only at temperatures ranging from 20 to 40 °C did free lipase show over 60% of relative activity. Although temperature profiles of immobilized biomolecules exhibited similarity in shape to those of free enzyme, significantly higher relative activityof those systems was observed over the whole analyzed temperature range. At temperatures ranging from 20 to 70 °C, over 60% of relative activity was noticed for all analyzed biocatalytic systems. Finally, similarly as analyzing pH effect on relative activity of immobilized lipase, slightly higher activity was noticed when MWCNTs/PDMS-100(40) material was used as support for enzyme immobilization.

#### 3.2.3. Thermal Stability of Free and Immobilized Lipase

Thermal stability of free and immobilized lipase was determined via samples incubation for 3 h at a temperature of 30 °C and atpH 7 (Figure 6). A relative activity decrease over incubation time for free and immobilized lipase was observed. However, the drop of catalytic properties was more pronounced for free enzyme, which retained less than 20% of its relative activity after 3 h of incubation. Significantly higher values of relative activity were noticed for biocatalytic systems produced. The decrease of catalytic activity of immobilized lipase was much slower as compared to the free counterpart; after 1 h and 3 h of incubation, immobilized enzyme showed around 40% and 50% higher values of relative activity, respectively. Finally, both biocatalytic systems obtained using MWCNTs/PDMS nanocomposites showed relative activity exceeding 80% after specific incubation periods.

In order to clearly present improvement of lipase stability and activity upon immobilization, enzyme inactivation constant (*k_D_*) and enzyme half-life (*t*_1/2_) were determined (Table 2). These parameters were calculated based on the linear regression slope from the above-presented Figure 6. Free lipase was characterized by *k_D_* = 0.01075 min^−1^ and a half-life of 64.74 min. Inactivation constant and enzyme half-life of immobilized lipase were improved. The most predominant increase of enzyme stability was noticed for lipase immobilized onto MWCNT/PDMS-100(40) material. Over a seven-fold lower inactivation constant (0.00126 min^−1^) and consequently over a seven-fold higher enzyme half-life (446.15 min) were observed for this particular biocatalytic system.

#### 3.2.4. Storage Stability and Reusability of Free and Immobilized Lipase

From apractical application point of view, storage stability and reusability are the crucial properties determining possible large-scale use of the immobilized enzymes. In this study storage stability of free and immobilized lipase was followed over 30 days, and reusability potential was examined over ten repeated reaction cycles (Figure 7).

It can be seen (Figure 7a) that storage stability of the lipase gradually decreased from the first day of storage; after 30 days it reached less than 20%. By contrast, storage stability of all tested biocatalytic systems with immobilized enzyme was improved significantly. Obtained biocatalytic systems showed over 90% relative activity after 6 days of storage and over 80% after 30 days. Further, lipase immobilized onto MWCNTs/PDMS nanocomposites showed over 90% activity retention at the end of the test.

Results of the reusability study (Figure 7b) showed that relative activity of the lipase immobilized onto MWCNTs/PDMS nanocomposites and P-MWCNTs remained almost unaltered for the first three reaction steps. Over the next experimental steps, catalytic activity decreased slightly. After ten cycles, relative activity of lipase immobilized onto P-MWCNTs reached 85%, whereas activity of lipase immobilized onto MWCNTs/PDMS nanocompositesattained over 90%.

#### 3.2.5. Effect of Solvents on the Immobilized Lipase

The effects of surfactant (Triton X-100) and salt solution (0.5 M NaCl) on catalytic activity of immobilized enzymes and stability of enzyme binding were determined by incubation of produced biocatalytic systems in the presence of the mentioned solutions over time. In Figure 8, it can be seen that relative activity of immobilized enzymes decreased gradually over first 6 h of incubation. After that time, all biocatalytic systems showed less than 50% and less than 30% of relative activity in the presence of Triton X-100 and NaCl, respectively. Further treatment of the biocatalysts withthe solvents did not result in such a pronounced drop in relative activity. After 24 h of incubation of lipase immobilized onto MWCNTs/PDMS nanocompositesin Triton X-100 and NaCl solution, analyzed samples retained over 40% and over 20% of relative activity, respectively.

## 4. Discussion

### 4.1. Analysis of Nanocomposites before Lipase Immobilization

The presented results demonstrate the changes in the textural/structural properties of MWCNTs after modification with polymer. The obtained data can be discussed also in terms of the absolute values of the surface area, *S*_BET_, which sharply decreased from 222 m^2^/g to 76–77 m^2^/g after PDMS (−100, −12,500) grafting (in the amount of 40 wt.%) onto carbon nanotube surfaces. That can be explained by two factors: reducing the content of MWCNTs in the resulting polymer composites after PDMS modification, as carbon nanotubes are responsible forsurface area, as well as increasing the size (diameter) of MWCNTs due to the polymer grafting onto carbon nanotubes. It is known that surface area is inversely proportional to the particle size of the dispersed phase. In general, the polymer adsorption leads to suppression of the values of *V*_p_, *V*_meso_, and *V*_macro_ because each long PDMS macromolecule can bind carbon nanotubes and aggregate themin more compacted structures, whichleads to adecrease in the volume of voids between MWCNTs [49].

It was found that the prepared nanocomposites are characterized by different graphitization degrees according to the data of Raman spectroscopy. A relatively lower A_G_/A_D_ ratio (about 0.9) for polymer nanocomposites indicates a low graphitization degree and shows that graphitic layers are semi-crystalline and possess many defects related to the D band due to introduction of new functional groups to carbon nanotube surfaces. The G’ peak appears at 2672 cm^−1^ as an overtone of the D band and is believed to originate from finite-size disordered structures of graphite (i.e., with the sp^2^C atoms) in the surface layers of the nanocomposites [50]. It would be interesting to check in futurework the results obtained here with respect to other polymers and other carbon-based fillers.

The results of FTIR analysis confirmed the carbon structure of the P-MWCNTs and indicatedthe presence of carboxylic groups on their surface that facilitate further MWCNTs modification. Upon PDMS adsorption onto MWCNTs, new signals are observed in the FTIR spectra of both modified samples, whichsuggests effective modification using poly(dimethylsiloxane) at various molecular weights. Nevertheless, the most important findings were made based on analysis of FTIR spectra of samples after lipase immobilization. The presence of signals characteristic for vibrations of amide I, amide II, and hydroxyl groups clearly indicate effective deposition of the enzyme onto the surface of both modified materials [51]. Moreover, the higher intensity of the signals ascribed for lipase structure, observed in FTIR spectrum of the system formed using MWCNTs modified with PDMS-100, indicate that a greater amount of the enzyme was immobilized, and immobilized biocatalysts retained higher catalytic properties [52].

### 4.2. Immobilized Lipase Characterization

Obtained data clearly showed that although pH and temperature profiles of free and immobilized lipase are similar, enzymes attached to P-MWCNTs and MWCNTs/PDMS nanocompositesshowed significantly higher relative activity over wider pH and temperature ranges as compared to free counterparts. Moreover, significant improvement of thermal stability and enzyme half-life of the lipase after immobilization was observed. A drop of the catalytic properties of the lipase in conditions different than optimal is related to the electrostatic repulsion of positively and negatively charged ionic groups in the enzyme structure and is also caused by thermal denaturation of the enzyme at harsh temperature conditions [53,54]. By contrast, immobilized lipase showed over 80% relative activity over wide pH (6–9) and temperature (30–60 °C) ranges. This might be explained by the fact that upon immobilization, an external backbone for the enzyme structure is provided due the formation of stable enzyme–support interactions, which stabilize enzyme structure and protect against biocatalyst denaturation at harsh reaction conditions [55]. Nevertheless, similarity in the pH and temperature profiles, and the presence of optima at the same conditions for free and immobilized lipases, as well asretention of high catalytic activity by produced systems indicate that immobilization did not significantly interfere with enzyme structure and its active site. It should be highlighted that among tested samples, the highest activity and tolerance to reaction conditions are ascribed to the lipase immobilized onto MWCNTs/PDMS-100(40) material. This is directly related to the fact that PDMS provides the hydrophobic nature of the surface and consequently the hydrophobic microenvironment for the immobilized lipase. In these conditions, lipase might undergo a phenomenon called interfacial activation, whichis based on opening of the polypeptide lid of the enzyme active site, leading to improvement of the activity of the immobilized enzyme [56,57]. Finally, significant enhancement of enzyme thermal stability (up to 50% higher relative activity, as compared to free enzyme) and reduction ofinactivation constant of immobilized enzymes are related to the fact that immobilization providesa protective environment for the enzyme molecules, whichreduces conformational changes of the enzyme structure in the presence of long-heat exposure. The advancement of using PDMS modified support for lipase immobilization was recently proved. Li et al. [58] modified silk fabric by amino-functional poly(dimethylsiloxane) (PDMS) and used it as a support for lipase from *Candida* sp. immobilization. It was shown that lipase activity and stability increased upon immobilization onto the hydrophobic surface. However, it was emphasized that the amount of the PDMS used affected catalytic properties of the immobilized enzyme. In another study, macroporous ZIF-8 was modified with PDMS in order to builda hydrophobic pore space for lipase from *Aspergillus niger* immobilization. Immobilized lipase showed improved stability and was found as an effective biocatalyst in the transesterification process in biodiesel production [59].

Determination of the storage stability and reusability of immobilized lipases is of crucial importance, as these parameters are key, determining possible practical application of the biocatalytic systemsproduced. All produced biocatalysts showed over 80% relative activity, even after 30 days of storage and 10 repeated uses, clearly indicating possible large-scale application. Such improvement of long-term stability and reusability is mainly related to the stabilization of the enzyme structure upon immobilization as well as the protective effects of both the MWCNTs support and the PDMS layer on the enzyme structure against inactivation over storage and reuse [60]. In the next part of the study, in order to determine the stability of the formed enzyme–support interactionsand the effects of various solvents on relative activity of immobilized enzymes, produced systems were incubated for 24 h in Triton X-100 and NaCl solutions. A significant drop in relative activity of immobilized lipase in the presence of both solvents might be explained mainly by two factors. Firstly, enzyme support interactions are based mainly on hydrogen and van der Waals interactions, which in the presence of Triton X-100 and NaCl lead to the partial elution of the enzyme from the support and decreased catalytic properties. Further, ionic strength might also affect catalytic properties of the immobilized enzyme by disturbing ionic interactions in the structure of the enzyme [61]. Partial elution of the enzyme and its inhibition by the reaction products are also explained by a slight decrease in the relative activity of the enzyme under repeated use. All of the above-mentioned facts negatively affect storage stability and reusability of the biocatalytic systems produced. Nevertheless, retention of over 80% activity after 30 days of storage, and 10 repeated catalytic cycles by the designed system, suggest that further study ofapplication of MWCNTs and PDMS modified MWCNTs as supportsfor enzyme immobilization are still required. In another study, Jamie et al. [62] immobilized lipase by covalent binding onto MWCNTs modified by n-2-hydroxysuccinimide/(1-ethyl-3-(3-dimethylaminopropyl)carbodiimide) (NHS/EDC) approach. Significant improvement of enzyme activity and operational stability was noticed, which is in agreement with the findings presented in this manuscript. Further, Khan et al. [63] immobilized lipase by adsorption onto MWCNTs treated with NHO_3_ and H_2_SO_4_. In this study, a protective effect of the support material was confirmed; however, it was highlighted that the initial concentration of the enzyme solution plays an important role in the final activity of immobilized lipase.

Recently, lipases of various origin were immobilized using a wide range of support materials, including sol–gel derived silica, zeolites, as well as synthetic polymers and biopolymers [64,65,66,67,68] (Table 3). In the presented studies, usually adsorption immobilization was applied resulting in production of biocatalytic systems characterized by retention of high catalytic activity and significant long-term stability and reusability. Further, application of obtained systems in hydrolysis reactions results in the attaining of over 90% process efficiency. In this context, lipase immobilized using biopolymers (modified chitin, spongin scaffolds), the application of which results in 100% transesterification efficiency, seems to be of particular importance [30,68]. Presented in this study approach, where MWCNTs modified with PDMS were used, results in production of a highly active biocatalytic system that retained 94% of its catalytic activity and over 90% of activity after 20 days of storage and 10 repeated catalytic cycles. High long-term stability and recycle potential of the obtained systems facilitates their potential in real condition applications, for instance in biodiesel production or in the pharmaceutical industry.

## 5. Conclusions

In the presented study, the fabrication of highly stable and active biocatalysts based on *Candida antarctica* lipase B (CALB) immobilized onto pristine and modified MWCNTs by poly(dimethylsiloxane) was reported. During material characterization, it was proved that the textural characteristics ofMWCNTs change after the modification with polymer, and that the prepared nanocomposites are characterized by a different graphitization degree, which results from, e.g., surface modification of carbon nanotubes with polymer—a lower graphitization degree; graphitic layers are semi-crystalline and possess many defects related to the introduction of new functional groups to carbon nanotube surfaces. Effective MWCNTs with poly(dimethylsiloxane) as well as enzyme loading were confirmed by bands present on FTIR spectra, characteristic for both modifier and biomolecule structures, which all together confirmed relative high potential of synthesized MWCNTs-based materials as a support for lipase immobilization. Enzyme loaded onto P-MWCNTs and MWCNTs/PDMS nanocomposites showed significantly higher relative activity over wider pH and temperature ranges as compared to free counterparts. Moreover, significant improvement of thermal stability and enzyme half-life of the lipase after immobilization was observed. It was confirmed that after immobilization, the external backbone for the enzyme structure was provided due the formation of stable enzyme–support interactions, which stabilize the enzyme structure and protect against biocatalyst denaturation under harsh reaction conditions. This fact suggests wide application potential of designed novel types of biocatalytic systems in various technological/biotechnological applications.

## Figures and Tables

**Figure 1 materials-14-02874-f001:**
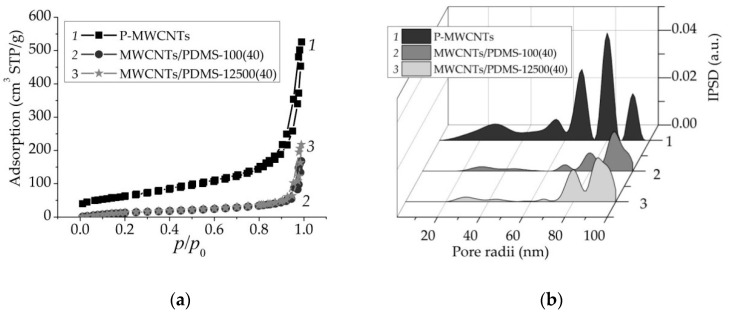
(**a**) Nitrogen adsorption–desorption isotherms and (**b**) incremental pore size distributions for P-MWCNTs (curve *1*) and MWCNTs/PDMS nanocomposites (curves *2*, *3*).

**Figure 2 materials-14-02874-f002:**
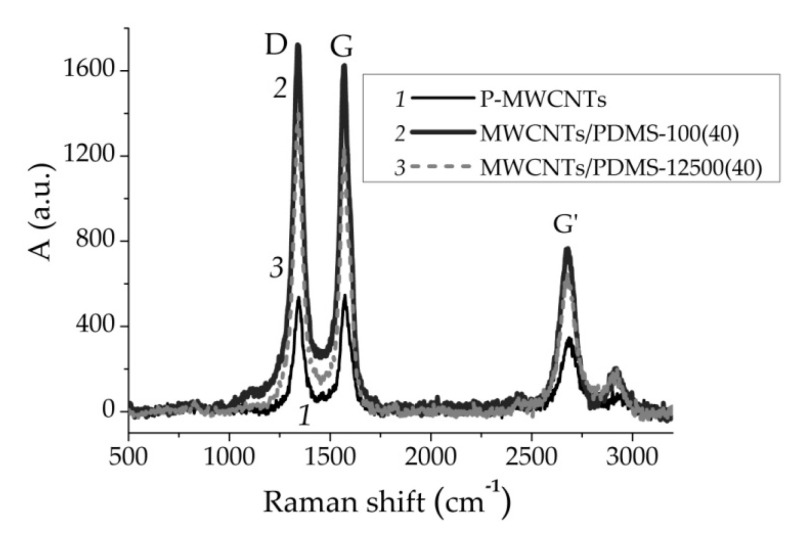
Raman spectra for P-MWCNTs (curve *1*) and MWCNTs/PDMS nanocomposites (curves *2*, *3*).

**Figure 3 materials-14-02874-f003:**
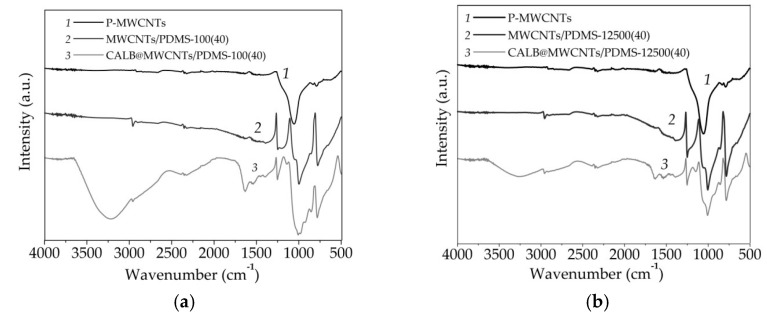
FTIR spectra of (**a**) P-MWCNTs, MWCNTs/PDMS-100, and (**b**) MWCNTs/PDMS-12500 nanocomposites, before and after lipase immobilization.

**Figure 4 materials-14-02874-f004:**
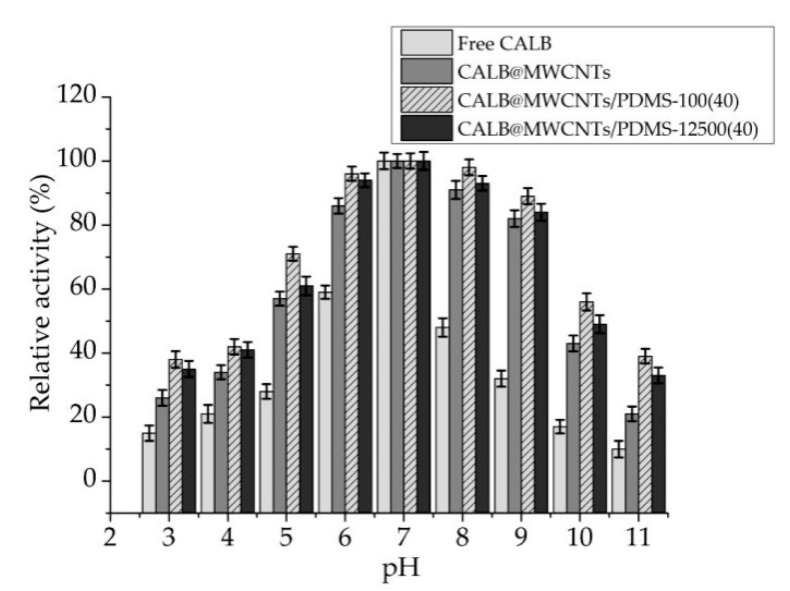
pH profiles of free lipase and enzymeimmobilized onto P-MWCNTs and MWCNTs/PDMS nanocomposites.

**Figure 5 materials-14-02874-f005:**
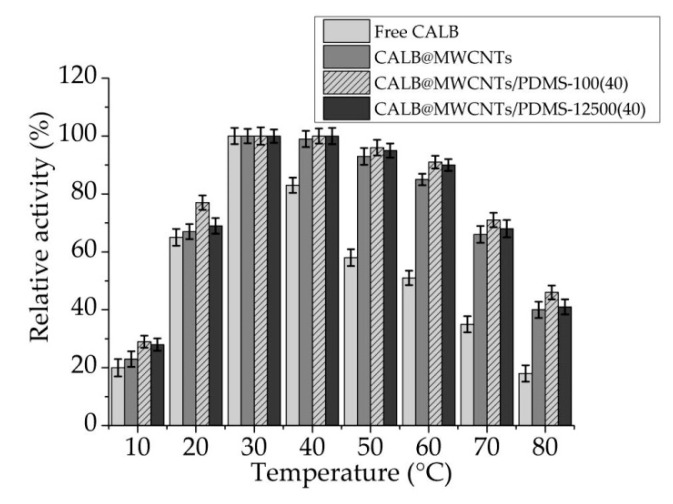
Temperature profiles of free lipase and enzyme immobilized onto P-MWCNTs and MWCNTs/PDMS nanocomposites.

**Figure 6 materials-14-02874-f006:**
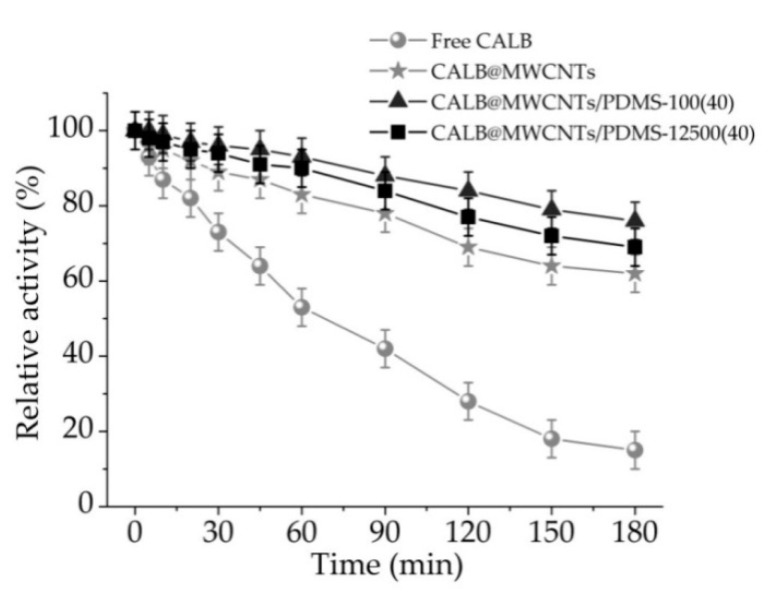
Thermal stability of free lipase and enzyme immobilized onto P-MWCNTs and MWCNTs/PDMS nanocomposites.

**Figure 7 materials-14-02874-f007:**
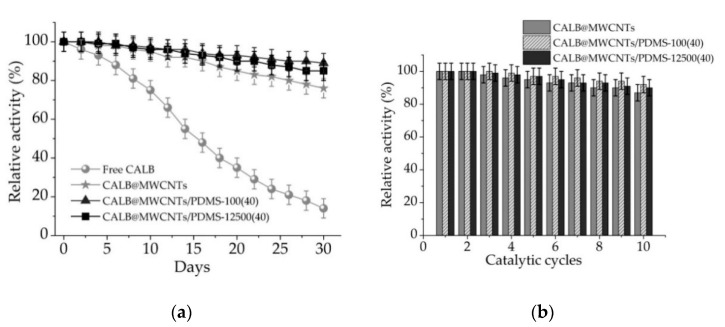
(**a**) Storage stability and (**b**) reusability of free lipase and enzyme immobilized onto P-MWCNTs and MWCNTs/PDMS nanocomposites.

**Figure 8 materials-14-02874-f008:**
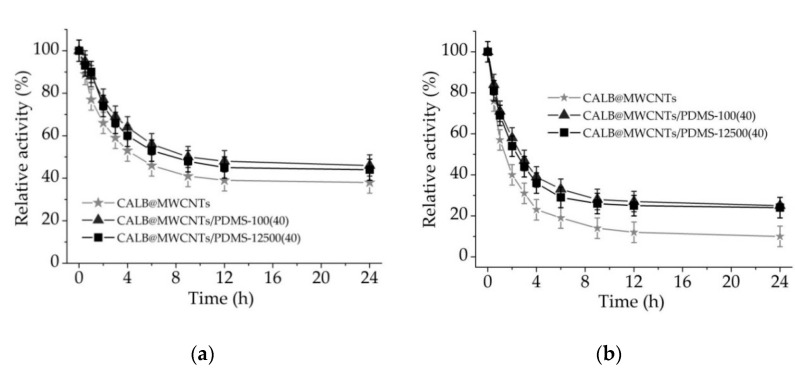
Effect of (**a**) 5% Triton X-100 solution and (**b**) 0.5 M NaCl solution on the relative activity of lipase immobilized onto P-MWCNTs and MWCNTs/PDMS nanocomposites.

**Table 1 materials-14-02874-t001:** Textural characteristics of P-MWCNTs and MWCNTs/PDMS nanocomposites.

Sample	*S*_BET_ (m^2^/g)	*S*_micro_ (m^2^/g)	*S*_meso_ (m^2^/g)	*S*_macro_ (m^2^/g)	*V*_micro_ (cm^3^/g)	*V*_meso_ (cm^3^/g)	*V*_macro_ (cm^3^/g)	*V*_p_ (cm^3^/g)	R_p,V_ (nm)
P-MWCNTs	222	74	134	14	0.039	0.418	0.357	0.814	23
MWCNTs/PDMS-100(40)	76	0	56	20	0	0.056	0.203	0.259	62
MWCNTs/PDMS-12500(40)	77	0	51	26	0	0.054	0.283	0.337	65

**Table 2 materials-14-02874-t002:** Inactivation constant and half-life of free lipase and enzyme immobilized onto P-MWCNTs and MWCNTs/PDMS nanocomposites.

Parameter	Free CALB	CALB@P-MWCNTs	CALB@MWCNTs/ PDMS-100(40)	CALB@MWCNTs/ PDMS-12500(40)
*k_D_* (min^−1^)	0.01075	0.00268	0.00156	0.00208
*t*_1/2_ (min)	64.74	259.70	446.15	334.61

**Table 3 materials-14-02874-t003:** Comparison of the most important parameters of lipase immobilized using various support materials. n.a.—not available.

Enzyme	Support	Type of Immobilization	Reusability	Storage Stability	Activity Retention	Process Efficiency	Ref.
Lipase from *Rhizomucor miehei*	Pure silica zeolites	Adsorption	60% after 4 catalytic cycles	n.a.	68%	93% of methyl myristate conversion	[64]
*Fusarium solanipisi* recombinant cutinase with high lipolytic activity	Zeolite	Adsorption	n.a.	89% after 45 days	74%	91% of trycaprylin transformation	[65]
Commercial lipases from *Rhizomucor miehei*	Polypropylene	Adsorption	85% after 8 catalytic cycles	n.a.	over 70%	90% of sunflower oil methanolysis	[66]
Lipase from *Rhizornucor rniehei*	Sol–gel silica	Entrapment	n.a.	75% after 20 days	86%	n.a.	[67]
Lipase B from *Candida antarctica*	*Hippospongiacommunis* spongin scaffolds	Adsorption	82% after 20 catalytic cycles	85% after 20 days	91%	100% of rapeseed oil methanolysis	[30]
Lipase B from *Candida antarctica*	Chitin modified by POSS * compounds	Adsorption	87% after 15 catalytic cycles	90% after 20 days	87%	100% of rapeseed oil methanolysis	[68]
Lipase B from *Candida antarctica*	MWCNTs modified by PDMS	Adsorption	91% after 10 catalytic cycles	90% after 20 days	94%	n.a.	this study

* POSS—polyoctahedralsilsesquioxanes.

## Data Availability

The datasets supporting the conclusions of this work are included within the article. Any raw data generated and/or analyzed in the present study are available from the corresponding author upon request.

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
