# Peer review of "Pristine and Poly(Dimethylsiloxane) Modified Multi-Walled Carbon Nanotubes as Supports for Lipase Immobilization"

_materials, 2021, doi:10.3390/ma14112874_

Round 1
Reviewer 1 Report
The present study dealt with fabrication of a higly stable and active biocatalysts based on Canadian antarctica lipase B immbilized onto prisitine and modified MWCNTs by poly(dimethylsiloxane). Nitrogen adsorption-desorption test, ATR-FTIR, Raman Spectroscopy, pH/temperature profiles, and thermal stability confirmed relative high potential of synthesized MWCNTs-based materials as a support for lipase immobiliation. Enzyme loaded onto PMWCNTs and MWCNTs/PDMS nanocomposites showed significantly higher relative activity over wider pH and temperature range as compared to free counterpart. Since it stabilizes enzyme structure and protect against biocatalysts denaturation at harsh conditions, it is deemed to have wide application potential of designed novel type of biocatalytic systems in various technological/biotechnological applications.
However, I checked spacing between words were not properly located in many sentences. I hope the authors to check whether this errors result from word-processing steps.
Author Response
Reviewer #1
The present study dealt with fabrication of a higly stable and active biocatalysts based on Canadian antarctica lipase B immbilized onto prisitine and modified MWCNTs by poly(dimethylsiloxane). Nitrogen adsorption-desorption test, ATR-FTIR, Raman Spectroscopy, pH/temperature profiles, and thermal stability confirmed relative high potential of synthesized MWCNTs-based materials as a support for lipase immobiliation. Enzyme loaded onto PMWCNTs and MWCNTs/PDMS nanocomposites showed significantly higher relative activity over wider pH and temperature range as compared to free counterpart. Since it stabilizes enzyme structure and protect against biocatalysts denaturation at harsh conditions, it is deemed to have wide application potential of designed novel type of biocatalytic systems in various technological/biotechnological applications.
Query 1: However, I checked spacing between words were not properly located in many sentences. I hope the authors to check whether this errors result from word-processing steps.
Answer 1: We would like to thank the Reviewer for this comment. The whole manuscript has been very carefully checked in order to remove spacing not properly located as well as to meet all of the Journal highest standards.
Reviewer 2 Report
The manuscript of Sulym and co-workers deals on the preparation and characterization of heterogeneous biocatalyst made by MWCNTs and lipase enzyme.
The work is quite interesting but the novelty is not adequately explained. In order to overcome this aspect, the comparison with other important supports and heterogeneous biocatalysts produced must be inserted. Please insert more literature on the argument and, in particular, make comparison among the materials reported in the following studies:
Support: zeolites – References: Catalysis Letters, 2008, 122(1-2), 43-52; J. Mol. Catal. B-Enzym., 1 (1996) 53–60;
Support: polymer – Reference: Enzym. Microb. Technol., 33 (2003) 97-1;
Support: sol-gel matrix– Reference: Appl. Biochem., 36 (2000) 181-186.
This comparison can help the authors to highlight the obtained results and the performance of their biocatalysts.
The catalysts characterization is well done, even if the biocatalyst activity is always measured by indirect methodologies. Testing the immobilized enzyme, by more cycles, should be interesting in a direct reaction, for example in the transesterification reaction to biodiesel production or in the hydrolysis reaction. Please, insert some test of this type, comparing them with the results of some of above mentioned published studies.
After this major integration, I think that the work is worthy of publication.
Author Response
Reviewer #2
The manuscript of Sulym and co-workers deals on the preparation and characterization of heterogeneous biocatalyst made by MWCNTs and lipase enzyme.
Query 1: The work is quite interesting but the novelty is not adequately explained. In order to overcome this aspect, the comparison with other important supports and heterogeneous biocatalysts produced must be inserted. Please insert more literature on the argument and, in particular, make comparison among the materials reported in the following studies:
Support: zeolites – References: Catalysis Letters, 2008, 122(1-2), 43-52;
- Mol. Catal. B-Enzym., 1 (1996) 53–60;
Support: polymer – Reference: Enzym. Microb. Technol., 33 (2003) 97-1;
Support: sol-gel matrix– Reference: Appl. Biochem., 36 (2000) 181-186.
This comparison can help the authors to highlight the obtained results and the performance of their biocatalysts.
Answer 1: We would like to thank the Reviewer for these suggestion. The novelty of the study has been highlighted in the Introduction section. We have also augmented the novelty of the obtained results by their comparison with previously published research concerning lipase immobilization onto various support materials. Proper paragraph and Table 3 has been added in the Results and Discussion section.
Query 2: The catalysts characterization is well done, even if the biocatalyst activity is always measured by indirect methodologies. Testing the immobilized enzyme, by more cycles, should be interesting in a direct reaction, for example in the transesterification reaction to biodiesel production or in the hydrolysis reaction. Please, insert some test of this type, comparing them with the results of some of above mentioned published studies.
Answer 2: We fully agree with the Reviewer that application of the obtained biocatalytic systems in the direct reaction at real process conditions would be the most suitable method for determination of their catalytic potential. However, due to pandemic situation and extremely short time for manuscript revision, we are unable to perform proper tests. Nevertheless, we thank the Reviewer for this remark, and in our next study concerning lipase immobilization, real test application will be properly planned and performed.
In the revised version of the manuscript we have added proper examples of the previously published study dealing with application of immobilized lipases for hydrolysis reactions (Table 3 in the revised manuscript). We have also highlighted the great potential of the presented in this study systems for real applications.
After this major integration, I think that the work is worthy of publication.
Reviewer 3 Report
This paper reports the preparation of pristine and two different molecular weight PDMS modified MWCNT nanomaterials, and CALB immobilized biocatalyst is further prepared based on the MWCNT nanomaterials. A wide range of characterization has been carried out, including the textural and structural properties, Raman spectrometry, and FTIR. For the bioactive immobilized enzyme catalyst, they investigated the immobilization efficiency, stability, and catalytic activity, and reusability. I think these results are interesting and I recommend acceptance, subject to the following, minor corrections:
1 In the introduction section, the authors focus on the discussion of MWCNT. For the related system such as single-walled carbon nanotubes and graphene, is there any research done with similar approaches? If yes, what’s the difference between the other systems and the MWCNT composite in this paper?
2 For the materials design section, can the author provide more information about the reason they use these two molecular weight PDMS, PDMS-100, and PDMS-12500? Is there any related literature for choosing these materials?
3 In the discussion section of nanocomposites analysis (3.1.1. Parameters of the Porous Structure), these two PDMS-modified materials have much lower surface area values (76 and 77) compared to the non-modified materials (222). Can the author provide some explanation for this result? Is this phenomenon also present in other similar systems?
Author Response
Reviewer #3
This paper reports the preparation of pristine and two different molecular weight PDMS modified MWCNT nanomaterials, and CALB immobilized biocatalyst is further prepared based on the MWCNT nanomaterials. A wide range of characterization has been carried out, including the textural and structural properties, Raman spectrometry, and FTIR. For the bioactive immobilized enzyme catalyst, they investigated the immobilization efficiency, stability, and catalytic activity, and reusability. I think these results are interesting and I recommend acceptance, subject to the following, minor corrections:
Query 1: In the introduction section, the authors focus on the discussion of MWCNT. For the related system such as single-walled carbon nanotubes and graphene, is there any research done with similar approaches? If yes, what’s the difference between the other systems and the MWCNT composite in this paper?
Answer 1:
A brief overview of studies about similar systems and reference to the literature data has been added to the Introduction section.
Query 2: For the materials design section, can the author provide more information about the reason they use these two molecular weight PDMS, PDMS-100, and PDMS-12500? Is there any related literature for choosing these materials?
Answer 2: Lipases are well-known, interracially active catalysts and exhibit their catalytic abilities at the interface between the organic phase containing hydrophobic substrates and aqueous phase, so can be activated at the hydrophobic–hydrophilic interface [Zhang,C.; Luo, S.; Chen,W. Talanta 2013, 113, 142–147. doi: 10.1016/j.talanta.2013.03.027]. Immobilization of lipase on the hydrophobic surface induces changes in protein conformation, and stabilization of lipase in its active conformation increases its stability and activity [Zahra Rastian, Abbas Ali Khodadadi, Zheng Guo, Farzaneh Vahabzadeh, Yadollah Mortazavi. Plasma Functionalized Multiwalled Carbon Nanotubes for Immobilization of Candida antarctica Lipase B: Production of Biodiesel from Methanolysis of Rapeseed Oil. Appl Biochem Biotechnol DOI 10.1007/s12010-015-1922-6]. Thus, the choice of hydrophobic polymer is critical. Among various soft matrices, poly(dimethylsiloxane) (PDMS) is a chemically inert, non-toxic, exhibit high elasticity, easy processing as well is environmentally-friendly silicon based organic polymer. PDMS is regularly used to prepare superhydrophobic materials [Wang, Y.; Gong, X. Special oleophobic and hydrophilic surfaces: approaches, mechanisms, and applications. J. Mater. Chem. A 2017, 5, 3759– 3773, DOI: 10.1039/c6ta10474f; Cao, C.; Ge, M.; Huang, J.; Li, S.; Deng, S.; Zhang, S.; Chen, Z.; Zhang, K.; Al-Deyab, S. S.; Lai, Y. Robust fluorine-free superhydrophobic PDMS-ormosil@fabrics for highly effective self-cleaning and efficient oil-water separation. J. Mater. Chem. A 2016, 4, 12179– 12187, DOI: 10.1039/c6ta04420d]. Previously, in our articles the study of hydrophobic properties for systems based on mixed oxide and PDMS with different viscosity (PDMS-200, PDMS-400, PDMS-1000) was presented and discussed. [Iryna Sulym, Panagiotis Klonos, Mykola Borysenko, Polycarpos Pissis, Vladimir M. Gun’ko. Dielectric and Thermal Studies of Segmental Dynamics in Silica/PDMS and Silica/Titania/PDMS Nanocomposites // J. APPL. POLYM. SCI. – 2014. – Vol. 131, Iss. 23. – P. 1236–1246. DOI: 10.1002/app.41154; P. Klonos,·G. Dapei,·I.Ya. Sulym,·S. Zidropoulos,·D. Sternik,·A. DeryÅ‚o–Marczewska,·M.V. Borysenko,·V.M. Gun'ko,·A. Kyritsis,·P. Pissis. Morphology and molecular dynamics investigation of PDMS adsorbed on titania nanoparticles: Effects of polymer molecular weight // European Polymer Journal. – 2016. – V. 74. P. 64–80; Iryna Sulym, Olena Goncharuk, Dariusz Sternik, Konrad Terpilowski, Anna Derylo-Marczewska, Mykola V. Borysenko, Vladimir M. Gun’ko. Nanooxide/Polymer Composites with Silica@PDMS and Ceria–Zirconia–Silica@PDMS: Textural, Morphological, and Hydrophilic/Hydrophobic Features // Nanoscale Res Lett (2017) 12: 152. doi:10.1186/s11671-017-1935-x]. It was found that in case of adsorption of PDMS-1000 onto oxide surfaces, we have obtained the highest values of the contact angle with water.
So in the current work, in order to establish the influence of PDMS chains length on the textural/structural properties of MWCNTs/PDMS polymer nanocomposites as well as lipase activity of produced materials, PDMS-100 and PDMS-12500 were used.
Query 3: In the discussion section of nanocomposites analysis (3.1.1. Parameters of the Porous Structure), these two PDMS-modified materials have much lower surface area values (76 and 77) compared to the non-modified materials (222). Can the author provide some explanation for this result? Is this phenomenon also present in other similar systems?
Answer 3: Surface area (SBET) is a parameter that characterize the total outer surface of particles of the dispersed system. In our case, there are two reasons for the decrease in the surface area of MWCNTs after grafting with PDMS. The surface area of polymer nanocomposites results mainly from SBET of MWCNTs (222 m2/g), because SBET of PDMS is close to 1 m2/g. After adsorption of both type of PDMS in the amount of 40 wt. % onto carbon nanotubes surface, SBET of MWCNTs/PDMS-100(40) and MWCNTs/PDMS-12500(40) polymer nanocomposites decreases up to 133 m2/g. But there is additional reduction of SBET values (up to 76 and 77 m2/). Thus, the second reason for these results is an increase in size of MWCNTs due to the polymer grafting onto carbon nanotubes. It is known, that surface area is inversely proportional to the particle size of the dispersed phase.
Results of our recent work [P. Klonos,·G. Dapei,·I.Ya. Sulym,·S. Zidropoulos,·D. Sternik,·A. DeryÅ‚o–Marczewska,·M.V. Borysenko,·V.M. Gun'ko,·A. Kyritsis,·P. Pissis. Morphology and molecular dynamics investigation of PDMS adsorbed on titania nanoparticles: Effects of polymer molecular weight // European Polymer Journal. – 2016. – V. 74. P. 64–80; Panagiotis Klonos, Iryna Y. Sulym, Dariusz Sternik, Pavlos Konstantinou, Olena V. Goncharuk, Anna DeryÅ‚o–Marczewska, Vladimir M. Gun'ko, Apostolos Kyritsis, Polycarpos Pissis. Morphology, crystallization and rigid amorphous fraction in PDMS adsorbed onto carbon nanotubes and graphite // Polymer. – 2018. V. 139. – P. 130–144] concerning PDMS grafting ontooxide nanoparticles, MWCNTs and graphite also showed reduction in surface area value.
A brief explanation of a decrease of surface area after adsorption of polymer onto MWCNTs surface has been added to the text - the section 4.1. Analysis of nanocomposites before lipase immobilization.
Round 2
Reviewer 2 Report
The manuscript has been sufficiently improved.